# Development of Shared Modeling and Simulation Environment for Sustainable e-Learning in the STEM Field

**Anatolijs Zabasta** [1,*], **Volodymyr Kazymyr** [2], **Oleksandr Drozd** [2], **Sammy Verslype** [3], **Ludovic Espeel** [3] **and Rasa Bruzgiene** [4]

1   Institute of Industrial Electronics and Electrical Engineering, Faculty of Electrical and Environmental Engineering, Riga Technical University, Azenes 12/1, LV1048 Riga, Latvia
2   Information and Computer Systems Department, Chernihiv Polytechnic National University, Shevchenko, 95, 14035 Chernihiv, Ukraine; vvkazymyr@stu.cn.ua (V.K.); alpdrozd@yahoo.com (O.D.)
3   Department of Electrical Engineering, Faculty of Engineering Technology, KU Leuven, Bruges Campus Spoorwegstraat 12, 8200 Bruges, Belgium; sammy.verslype@kuleuven.be (S.V.); ludovic.espeel@kuleuven.be (L.E.)
4   Department of Computer Sciences, Kaunas University of Technology, Studentu St. 50-211 kab., LT-51368 Kaunas, Lithuania; rasa.bruzgiene@ktu.lt
*   Correspondence: anatolijs.zabasta@rtu.lv; Tel.: +371-29-232-872

**Abstract:** A novel educational platform called the Shared Modeling and Simulation Environment (SMSE) was initiated in the ERASMUS+ "CybPhys" project. It integrates the Jupyter platform, including Jupyter Notebooks, with the Moodle Learning Management System for e-learning in the STEM field. This novel platform enhances e-learning by combining the content of training courses in Moodle with practical programming provided Jupyter's capabilities to create virtual labs. A survey was conducted in the spring of 2023 among bachelor's and master's students at Chernihiv Polytechnic National University, which aimed at gathering feedback from students using SMSE in their e-learning process. Two student groups were involved: one of them consisted of students using SMSE in both the autumn and spring semesters and another represented the students starting with SMSE in the spring semester. The survey, based on the adjusted Technology Acceptance Model (TAM), was conducted to understand the acceptance of this e-learning approach. A comprehensive analysis of students' answers based on the TAM approach revealed the inner process of the transformation of students' perceptions during the acquisition of the SMSE platform. Our research demonstrates that SMSE effectively merges Moodle's online learning capabilities with Jupyter Notebooks, providing a flexible and interactive learning experience for both in-class and remote students. It provides a web-based, multifunctional e-learning environment that combines a variety of tools and technologies, giving students the possibility to be involved in all sorts of teaching activities related to STEM education, such as theoretical knowledge, exercises, simulations and calculations, through the use of one single online environment.

**Keywords:** emerging technologies in education; sustainability; e-learning; Moodle; Jupyter Notebook; shared modeling and simulation environment; Technology Acceptance Model

## 1. Introduction

Education can be considered one of the most effective ways to contribute to the implementation of sustainability [1]. However, training new generations in the fulfilment of the Sustainable Development Goals (SDGs) of the 2030 agenda is one of the challenges in education today, especially regarding online learning. The Director of the Sustainable Development Solutions Network (SDSN), Jeffrey D. Sachs, underlined the role of higher education: "Universities around the world should be at the forefront in helping society find technical solutions to achieve these goals" [2].

One of the most effective ways to achieve these goals is the implementation of STEM education [3] in the practice of pedagogical activities. Even though STEM education as an educational methodology appeared in the second half of the last century, it is now gaining special importance in responding to the challenges of the 21st century [4]. The integration of ICT into STEM education represents a pivotal evolution, enabling an enriched, interactive educational experience that aligns with the dynamic demands of modern society. This is facilitated, first of all, by the wide implementation of ICT in the educational process.

The growing application of new Information and Communication Technologies (ICT) in education practices has changed the nature of the teaching–learning environment and has significantly contributed to sustainable development [5]. Moreover, the adoption of ICT and online learning platforms heralds a transformative shift in education, offering new pedagogical possibilities and enhancing access to quality education across diverse contexts. Simultaneously, remote online learning has become a cornerstone of modern educational processes. It facilitates connectivity among students and between students and teaching professors [6]. It makes the educational process more resilient to disruptions originating from unforeseen circumstances like pandemics, political and economic instabilities, and even war.

At the same time, according to [7], despite considerable research on sustainability in education, there is a lack of studies that systematize the innovative proposals that can meet the challenges of educational sustainability, including ensuring inclusive, equitable and quality education, and promote lifelong learning opportunities for all. This also applies to the improvement of STEM education, which at the current stage needs modern ICT tools and new methods of implementation.

## 2. Literature Review

In the modern sense, a STEM education, which combines science, technology, engineering and mathematics, can be implemented according to three approaches [4]: the SILO-approach, where students have little opportunities to learn by doing (rather, they are taught to know); the Embedded approach, where students learn through understanding and application, but they cannot associate the embedded content to the context of the lesson; and the Integrated approach, which eliminates boundaries between STEM components. The last approach is certainly the best, but it requires pedagogical training for the teachers and special ICT tools.

Furthering the discourse on integrating educational technology, the importance of cultivating digital literacy among both educators and learners is emphasized. Work [8] discusses the development of sustainable technological tools, the best practices for making training accessible to university faculty members through ICTs, and the creation of virtual tools and courses to enhance education quality. The adoption of ICTs in academic settings not only improves pedagogical approaches but also fortifies educational systems against future disruptions. Moreover, the emphasis on digital transformation within universities highlights the need for ongoing professional development and adaptation strategies among faculty members to effectively utilize digital tools for enhanced learning outcomes. Research [9] explores the relationship between university professors' adaptation to the widespread use of ICTs and educational digitalization. This comprehensive approach to ICT integration in education signifies a shift towards more inclusive, flexible, and student-centered learning environments, enabled by the strategic deployment of digital technologies.

A novel e-learning platform, named Digital Brick, intended to enhance the students' experience in obtaining formal certifications of their competencies, is depicted in [10], which applies machine learning algorithms to provide students with personalized recommendations of online learning content. A work [11] describes the experience of using virtual laboratories (VL) in tertiary education when two types of VL are used: the first is simulation-type VL, where the experiment modeling is based on mathematical equations,

and the second is real-time remote conduction of experiments on the experimental set-up; however, only one person can experiment at a time.

Personalized or Precision Education (PE) considers the integration of multimodal technologies (e.g., MOOCs, serious games, Artificial Intelligence, learning management systems, etc.) to tailor individuals' learning experiences based on their preferences and needs. A review of emerging multimodal technologies to evaluate their impact on personalized education is depicted by [12].

As for the support of online learning, Moodle is the most widespread Learning Management System (LMS), having multiple options for course creation, management and grading. This platform allows students to submit their practical assignments but not complete them in an online environment, hence requiring every participant to set up a personal working environment on their machines. This not only introduces unnecessary complexity for the students and teaching staff but also causes complications (e.g., the necessity of licensing) or even prevents the students from completing the course due to inadequate hardware [13].

However, such issues can be resolved with relative ease by deploying an institution-wide virtualized solution that would provide users with on-demand, pre-packaged workspaces that could be accessed individually [14]. Additionally, this approach also ensures that all course participants have access to a homogeneous educational and working environment, thereby preventing the infamous "It works on my computer" issue from occurring. While the desired result may be archived in multiple ways, the Jupyter project [15] presents the solution with benefits such as scalability, modular design, a mature and rich ecosystem, a web-based user-facing interface and the ability to combine lecture material and runnable examples in a single Jupyter Notebook. So, the combination of Moodle and Jupyter platforms in one integrated technological solution would create the prerequisites for reaching the level of the Digital Learning Ecosystem (DLE), which, according to [16], is the most high-quality and effective form of teaching and learning in a period of instability associated with pandemics such as COVID-19, but also military conflicts, one might add.

As for the implementation forms of integration Moodle and Jupyter within the framework of an Integrated approach in STEM, it may be the argument presented in [17] that models and modeling could serve as a bridge between the STEM subjects in educational practice. Models range from simple conceptual diagrams to advanced mathematical models, algorithms and program code. Therefore, the competencies needed to create, use and apply models are necessary for gaining an in-depth understanding of scientific practice, technological and engineering design, and mathematical tools.

According to a survey [18], the majority of students had a positive attitude to conducting remote examinations and even considered such assessment methods as convenient and satisfactory. While e-learning is often seen as limited social interaction with insufficient social presence and unsynchronized communication, it certainly offers numerous benefits for students.

Thus, each of the aforementioned works provides a solution in one specific direction, ranging from providing online systems to distribute courses, using software approaches to aid with simulation and modeling, providing an environment to work in remote laboratories, etc. However, none of the solutions combines multiple education-oriented technologies to provide both the student as well as the teacher with a multifunctional e-learning environment. Ideally, students and teachers are provided with an e-learning environment that allows for a wide range of education-related aspects, such as lecturing, providing courses, making exercises, simulating and modeling systems, getting real-time feedback, using multiple programming and simulating languages, etc.

The purpose of this article is to highlight the main features of the novel online education platform as the implementation of an Integrated approach in STEM which allows for a wide range of education-related aspects, such as: lecturing, conducting courses, performing exercises, simulating and modeling systems, receiving feedback, and the use of many programming and modeling languages. The developed platform named as the

Shared Modeling and Simulation Environment (SMSE) attempts to answer this question by creating a web-based multifunctional e-learning environment for modeling and simulation that combines a variety of tools and technologies. The ultimate goal of this research is to identify the perceptions of students about the introduction of the SMSE platform in the e-learning process by applying the Technology Acceptance Model [19] approach.

The rest of the paper is organized as follows. In Section 3, we have provided a background on the study materials and a brief overview of the research methods. Section 4.1 describes the architecture of the Shared Modeling and Simulation Environment. Sections 4.2 and 4.3 depict the technical solution, and Section 4.4 describes the technical aspects of the e-learning process. In Section 5, we describe the details of the survey of two groups of students, analyze the students' questionnaires and discuss the impact of the novel e-learning platform on the study process. We make conclusions in Section 6 about the advantages and restrictions of SMSE and discuss plans for the further development of the platform.

## 3. Materials and Methods

The realization of the defined goal was undertaken within the framework of the ERASMUS+ Capacity Building in the Higher Education project 609557-EPP-1-2019-1-LV-EPPKA2-CBHE-JP "Development of practice-oriented student-centered education in the field of modeling cyber-physical systems" (CybPhys) [20,21], which was implemented in 2019–2023. Since the participants of the project were three Ukrainian universities (Chernihiv Polytechnic National University (CPNU), Kharkiv National Automobile and Road University (KHNAHU) and Kryvyi Rih National University (KNU)), the strengthening of distance learning and teaching methods in STEM became an urgent task, considering both the impact of the COVID-19 pandemic and Russia's military aggression, since a large number of students and teachers were forced to become displaced persons.

The competencies of Ukrainian partners were complemented by the significant experience of European colleagues from the Riga Technical University (RTU), KU Leuven University from Bruges, and the University of Cyprus (UCY) in the development of digital learning methods. As a result, a new Shared Modeling and Simulation Environment was created based on the ICT infrastructure of CPNU. Having remote access to SMSE, the other partner universities created one remote lab for each partner for testing and staff training.

The development of the SMSE was preceded by the study of the experience of using Jupyter Notebooks in pedagogical practice [22,23], the organization of computer-cognitive laboratories [24] and the creation of the interface of virtual JupyterLabs [25]. Considering the already existing practice of using the Moodle LMS, the issue of the integration of the above-mentioned features of the Jupiter platform into the existing and created educational content was successfully resolved. Further, thanks to the efforts of all project partners in testing, the use of the SMSE in the educational process became a prototype of an educational ecosystem for practical modeling of cyber-physical systems for innovative physical-mathematical and engineering topics [21]. Integration with Moodle contributed to the transformation of the primary focus of the project into modeling tasks for other educational courses studied at universities. This became a good opportunity to consider SMSE as a general approach to distance learning within the framework of the future DLE.

Currently, the use of SMSE at CPNU only covers three bachelor's, two master's, and four postgraduate courses. Thanks to the use of SMSE, teachers have the opportunity to create educational materials for laboratory works with the support of a wide range of software execution environments, and students can perform these works online in their virtual laboratories. Moreover, the entire process of joining SMSE takes place directly through Moodle, which enables authorized access to centralized storage and the use of course materials. This not only contributed to the activation of the SMSE among partners, but also prompted research on the efficiency of remote distributed laboratories to support training in the field of computer science and information technology.

At the end of the spring semester of 2023 we surveyed students, aiming to receive feedback on their perceptions about the application of the SMSE in the e-learning process. The survey was implemented in the form of an anonymous questionnaire and answered by the participants. The survey included a set of questions, where participants were asked to rate, according to their perceptions, the relevance of each of the following evaluation criteria. A Likert scale from 1 (not relevant) to 5 (very relevant) was used. The distance between each scale point is assumed to be equal, where "3" represents the neutral value on this scale.

We arranged a survey among two populations of students of CPNU: the first one represented bachelor's students in the third-year of study in computer engineering who acquired the SMSE and regularly used this platform in an e-learning form of education during the autumn and spring semesters; the second one represented bachelor's students in the second-year of study in computer engineering who had just started the acquisition of the course material using the SMSE in the spring semester. Therefore, both populations of students were trained using the e-learning platform, but the first group had more time to obtain an affinity with the SMSE approach. The anonymous survey of the students was performed using Moodle possibilities. We received twelve valid survey forms from students of the first population, which we called Group 1 (experienced students), and also twelve forms from the students of the second population, which we called Group 2 (less experienced students).

In the survey, 22 questions were aligned to the TAM questionnaire [26,27]. The questionnaire was focused on the main factors that encourage the acceptance of the suggested e-learning approach, which assumes the application of the SMSE, such as Perceived Ease of Use (PE), Perceived Usefulness (PU), Attitude (AT), Behavioral Intention (BI), E-learning Self-Efficacy (SE), Subjective Norm (SN), and System Accessibility (SA).

According to TAM, one's actual use of a technology system is influenced directly or indirectly by the user's behavioral intentions, attitude, perceived usefulness of the system, and perceived ease of the system [28]. TAM has evolved, extending the original model to explain perceived usefulness and usage intentions including social influence (subjective norm, voluntariness, and image), cognitive instrumental processes and experience.

We extended the model applied in [27] by a group of questions, which we called "In-depth acceptance of the new SMSE platform". Therefore, we extended the questionnaire by several new questions, 17–22, in addition to the proposed one in [27], aiming to obtain a more comprehensive view of students' perception of the SMSE application in the e-learning process (see Figure 1). Also, we asked the students to provide comments and suggestions about the training methods that might be used for further improvements.

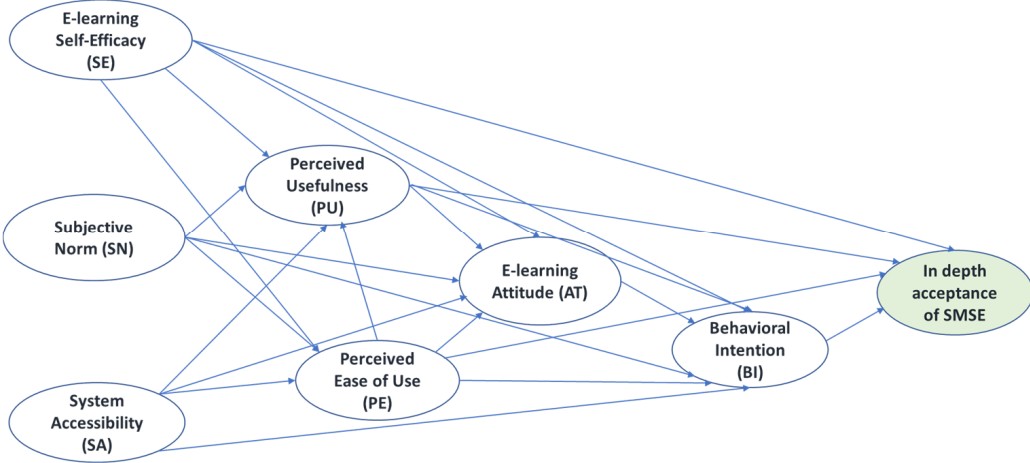

**Figure 1.** Extended TAM structural model.

## 4. Technological Base of SMSE Platform

*4.1. Architecture of the Shared Modeling and Simulation Environment*

The SMSE combines several well-known technologies and software products. These constituent parts can be used separately, but the best effect on learning is achieved precisely when the parts interact in one integrated structure. The SMSE has three main architectural components as shown in Figure 2.

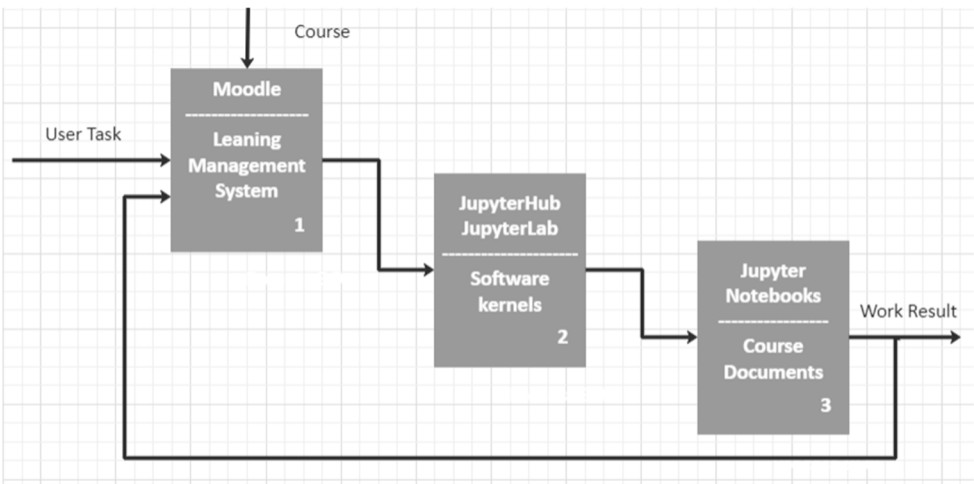

**Figure 2.** SMSE architecture.

Learning Management System. The SMSE uses the Moodle Learning Management System (LMS) to organize training courses, which require remote laboratories with pre-installed software. Through Moodle, users receive access to educational materials and other SMSE software components. At the same time, Moodle provides management of user accounts, including their authentication and authorization, through the regulation of access rights. All other SMSE components synchronize user information with Moodle. Personal user IDs generated during their authorization in Moodle are active throughout the entire process of using personal virtual laboratories.

Software kernels. The task of this architectural component is to create a virtual laboratory for the user and support the implementation of the practical component of the training course. As a rule, practical tasks are presented in the form of laboratory work that is performed in a certain software environment. To form such an environment, a basic server is selected. It runs a set of software kernels by the needs of the course. Users have the opportunity to perform laboratory work developed in a combined software environment with the support of different programming languages, due to the number of software kernels involved. The implementation of this component in SMSE was carried out using the tools of the Jupyter platform, such as JupyterLab 3.4.3, JupyterHub 3.0.0 and Jupyter Notebook with Python 3.10.5 [29]. JupyterLab implements the user's virtual laboratory for which JupyterHub runs the required server with a set of software kernels and provides users with course documents in Jupyter Notebook format. The scheme of data flows in the SMSE is shown in Figure 3.

Course Documents. This component contains the texts of laboratory works with code in programming languages, which are executed using the corresponding software kernels. Since the basic format of laboratory works is Jupyter Notebook, this document combines both textual and graphical information, as well as mathematical formulas, multimedia and links to external resources. But most importantly, Jupyter Notebook is able to include examples of program code that can be executed by running it directly from this document. Jupyter Notebook is an example of an interactive training course. The SMSE ensures the implementation of this course in the user's personal virtual laboratory. An example of a document in the Jupyter Notebook format is shown in Figure 4.

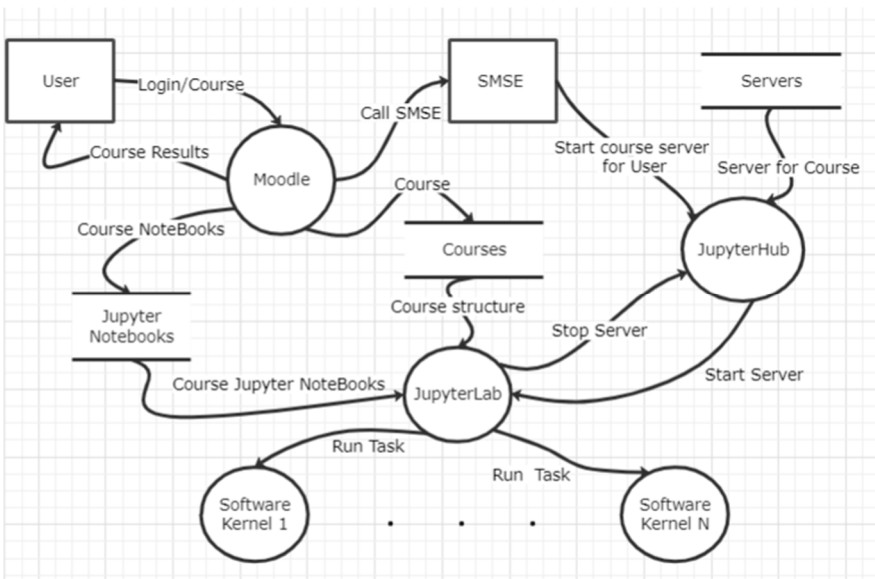

**Figure 3.** The scheme of data flows in the SMSE.

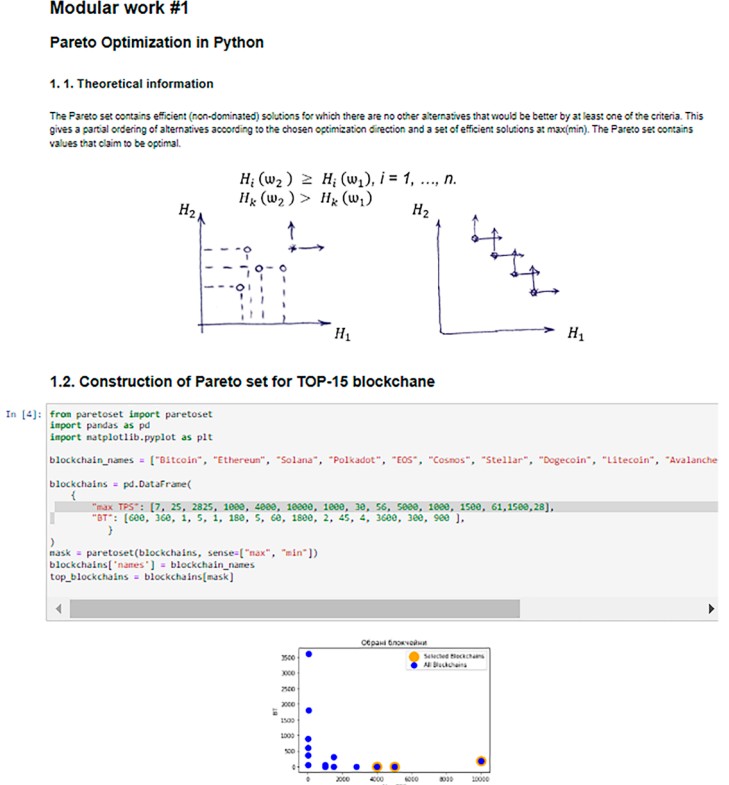

**Figure 4.** Example of Jupyter Notebook document.

### 4.2. Virtual Servers

In SMSE, Jupyter Notebook documents are launched for execution on the JupyterLab virtual server, which runs the software kernels [30] and provides additional tools for the execution of program code directly from Jupyter Notebook documents. To support multi-language examples of code within a single course, the SMSE uses Docker technology [31] to run on virtual server-separated resources instead of directly using the virtual machines on the operating system. The scheme of using Docker tools in SMSE is presented in Figure 5.

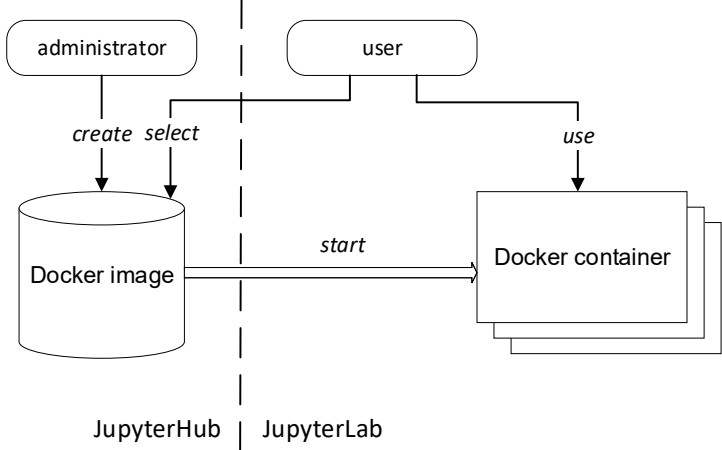

**Figure 5.** The scheme of using Docker tools in SMSE.

The SMSE administrator creates a Docker image as a virtual server for one course or a whole pool of courses, relying on basic templates or generating by themselves. Each Docker image contains everything needed to run the required runtime environment in the form of Docker containers.

When a user starts a selected virtual server in JupyterHub [32], the Docker-Engine creates for this user their own JupyterLab with a virtual server in the form of a Docker container, which is a copy of the corresponding Docker image. In JupyterLab, in addition to the general settings of the Docker container, the user can access the course files and their working directory through JupyterHub. The user's virtual server remains active until it is stopped by the user. Depending on location, the user's files may be deleted when the server is stopped, but files saved in the user's folders will not be deleted; they will be constantly available from the user's JupyterLab.

An important addition of JupyterLab is the ability to install additional programs on the virtual server by the user themselves. But these programs work only when the virtual server is running. Deleting temporary user data and additionally installed programs when stopping the virtual server can be considered a positive moment for saving physical server resources.

*4.3. Data Migration between Moodle and the SMSE Virtual Server*

Two types of data are transferred between Moodle and SMSE: user authentication data and Jupyter Notebook files. Pass-through authentication is performed in Moodle with the Learning Tools Interoperability (LTI) tool. To transfer course files, a specially developed Middleman Service was used. Its main tasks include:

- Syntactic analysis of the Moodle course structure to obtain a layout of course sections and a list of downloadable files;
- Creating a Moodle course archive for transferring course data to JupyterLab;
- Temporary storage of the course archive on the Git server.

Using the Middleman Service and an intermediate Git server enables a reduction in traffic between Moodle and JupyterLab. Files from the Git server are automatically fetched when the user's virtual server is started in JupyterHub. The scheme of the course file migration in the SMSE is shown in Figure 6.

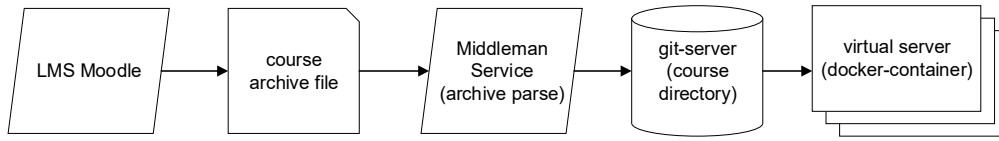

**Figure 6.** The scheme of course files migration in SMSE.

### 4.4. Organization of Online Training with SMSE

The main task of the SMSE is to support, in an online mode, the individual work of students with the course materials (lectures, practical or laboratory works) stored in Moodle. This possibility is provided by the use of remote laboratories with pre-installed software kernels that are created after running the virtual server related to a course. As actors in this process are both a teacher and a student, their use case diagrams are presented in Figures 7 and 8, respectively.

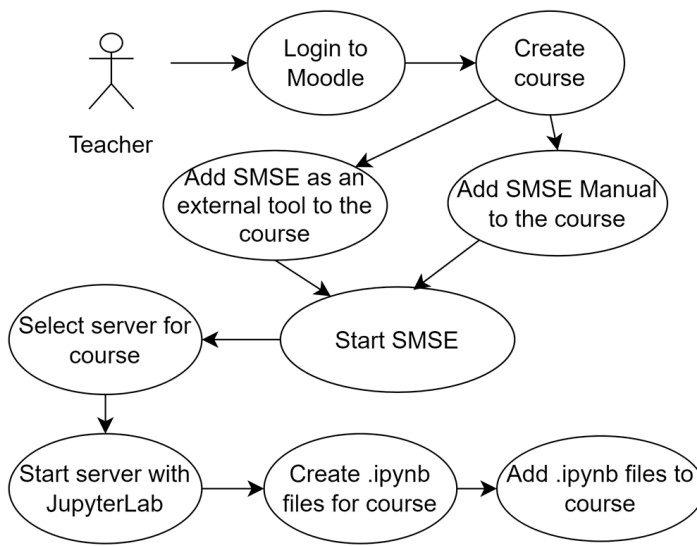

**Figure 7.** The teacher uses a case diagram.

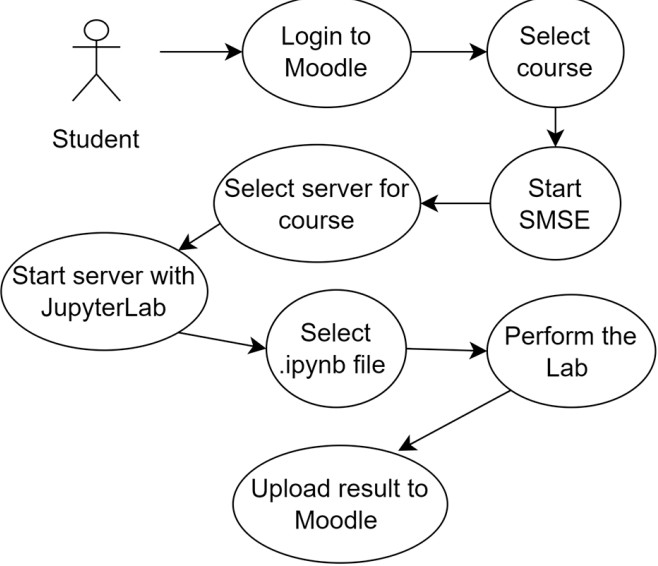

**Figure 8.** Student use case diagram.

The accepted form of educational material presentation in the SMSE is the Jupyter Notebook, with code examples in programming languages supported by virtual servers. Server images are created by the SMSE administrator at the teacher's prior request. The set of software kernels necessary for studying the course is formed by the teacher. Also, the teacher can use servers from the list of available servers previously created by the SMSE administrator. The SMSE page for selecting the desired server image is shown in Figure 9.

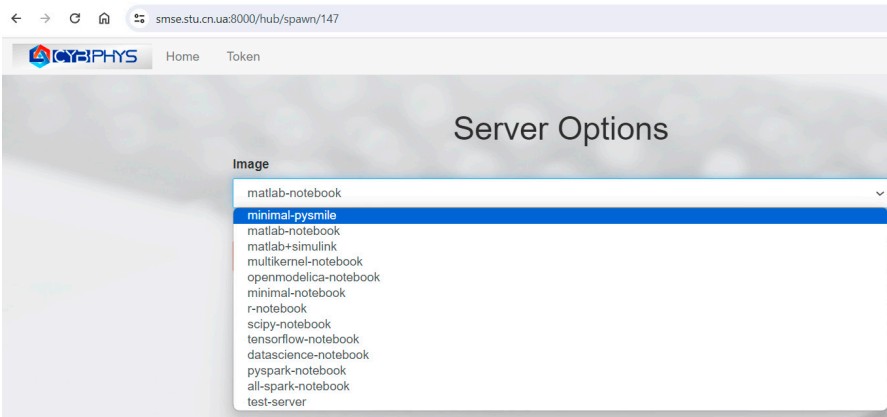

**Figure 9.** The SMSE page for selecting the server image.

The current implementation of SMSE includes 12 images of virtual servers that support various modeling environments, such as Matlab r2022a (including a variant with Simulink) and OpenModelica 1.22.1, as well as programming Java, R, Octave, Julia (in case of multi-kernel notebook) languages and a whole set of environments for Python with installed libraries. It should be noted that every server on the list supports Python by default, including the minimal-notebook, which is limited to Python only.

After starting the selected server, the course materials, represented in the form of Jupyter Notebook, are automatically loaded into the user's JupyterLab in the appropriate sections of the course. Also, all the software kernels of the running server become available in JupyterLab. Combining cells with text information in Markdown format, formulas in LaTex format and code in Code format, the teacher creates the educational content of the corresponding lesson as a Jupyter Notebook document. The code examples can be run directly from this document by previously selecting the desired kernel using the menu. The general view of the JupyterLab page of the multi-kernel notebook server for the Computer Modeling course is shown in Figure 10.

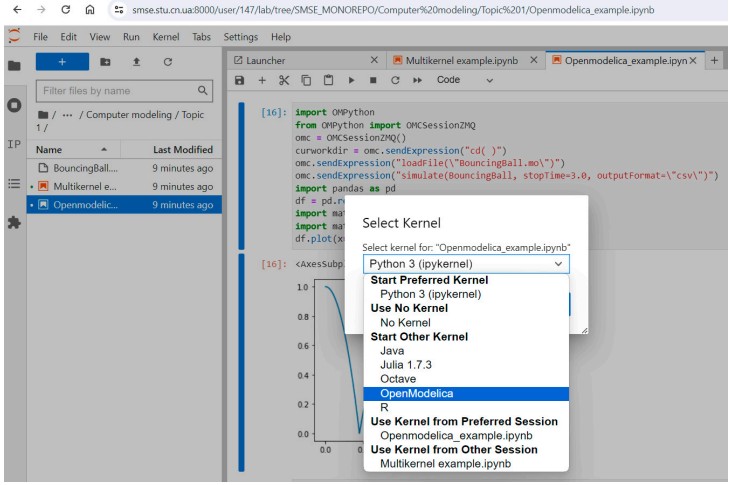

**Figure 10.** The general view of the JupyterLab page.

The created Jupyter Notebook document can be saved in the working directory of the teacher's JupyterLab and then uploaded to Moodle in the appropriate section of the course.

The student's actions in using SMSE are rather similar to the teacher's actions. When launching SMSE from the Moodle course, the student selects the recommended server, and after starting their JupyterLab, they can open the required Jupyter Notebook for studying, editing and saving the result of the processing task in the Jupyter Notebook format in their working directory for further uploading to Moodle in the corresponding section with the

results of the completed work. It should be noted that the saving of all users' achievements in the working directory of JupyterLab for further use is because each time JupyterLab is started, the course files are updated to their current state in Moodle. The general view of the Moodle course with built-in SMSE resources is shown in Figure 11.

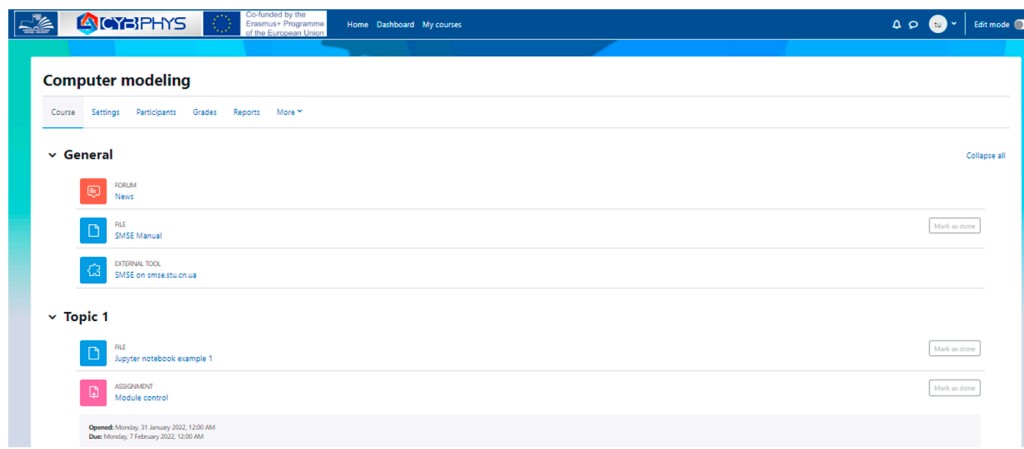

**Figure 11.** The general view of the Moodle course with built-in SMSE resources.

The integration of SMSE with Moodle not only enables flexible management of user accounts and course content, but also allows the use of other Moodle options for the assessment of students' results and monitoring their progress. Also, it allows the use of questionnaires to obtain students' feedback for improving the learning environment. Figure 12 shows an example of filling out a questionnaire regarding the usefulness of SMSE in the educational process.

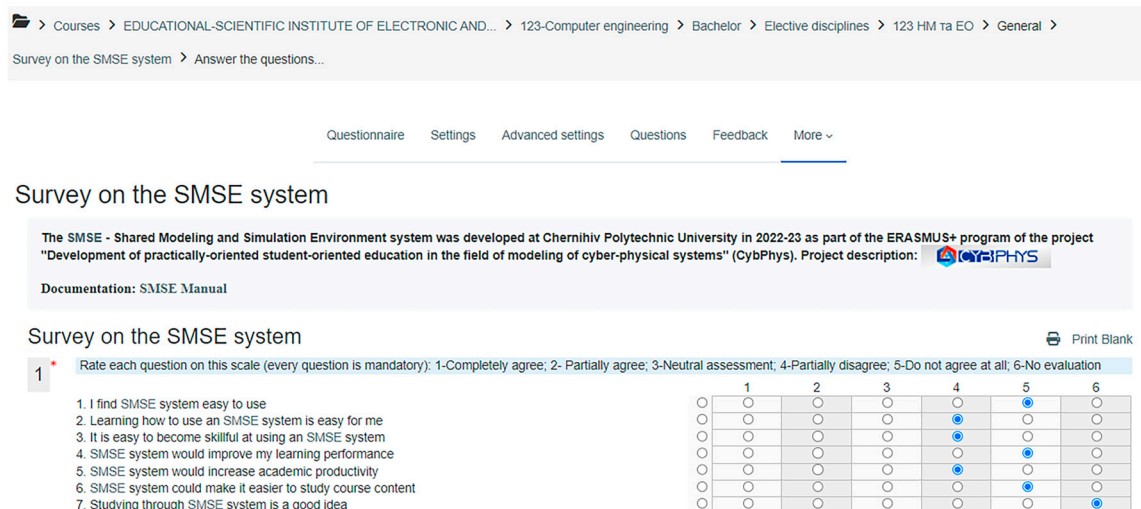

**Figure 12.** An example of a questionnaire regarding the usefulness of SMSE.

## 5. Survey of Two Groups of Students about the Acquisition of the SMSE Platform

### 5.1. Analysis of Results of the Student's Answers

Twelve respondents in each group filled out the questionnaires that were considered valid. To develop confidence in the comparison of the perceptions of two groups of students (see Section 3, Materials and Methods), we used the hypotheses which determined whether there were significant differences between the respondent's perceptions. The null hypothesis, $H_0$, equaled $\mu_1 - \mu_2 = 0$, i.e., there was no difference between the perceptions of both groups. The alternative hypotheses are the following:

**H$_{A1}$:** *μ$_1$ > μ$_2$ if we believe the mean for Group 1 is greater than the mean for Group 2.*

**H$_{A2}$:** *μ$_1$ < μ$_2$ if we believe the mean for Group 1 is less than the mean for Group 2.*

Where μ$_1$ and μ$_2$ are the means of the evaluation marks of Group 1 and Group 2 using the Likert scale.

We calculated descriptive statistics related to the survey of both groups. Due to the lack of space, we omit the calculations and tables of the descriptive statistics. Since the skewness of both samples is 0.623 and 0.508, we admit that the data can be considered normally distributed; therefore, the student's two-sample *t*-test could be applied [33]. In addition, the two variances are relatively similar (0.0939 and 0.0402), which also implies the use of the *t*-test: the Two-Sample Assuming Equal Variances data analysis tool was used to test the null hypothesis [34].

Since t$_{obs}$ = 1.84399 < 2.018 = t$_{crit}$ (or *p*-value = 0.0722 > 0.05 = α), we retain the null hypothesis, i.e., we are 95% confident that any difference between the two groups is due to chance.

In addition, to be sure, we also applied the *t*-test, Two-Sample Assuming Unequal Variances. We received a very similar result: t$_{obs}$ = 1.84399 < 2.028= t$_{crit}$ (or *p*-value = 0.0734 > 0.05 = α); therefore, we cannot reject the null hypothesis (for the two-tailed test). It means that there are no significant differences between the respondent's perceptions in both groups.

As mentioned in Section 3, "Materials and Methods", we asked 22 questions aligned to the TAM questionnaire aiming to identify the main factors that encourage the acceptance of the suggested e-learning approach, which assumes the application of the SMSE. Figure 13 depicts the answers of students from Group 1 who regularly used the SMSE (a histogram of cumulative results). On the other hand, Figure 14 depicts the answers of students from Group 2, who recently started to use the SMSE (a histogram of cumulative results).

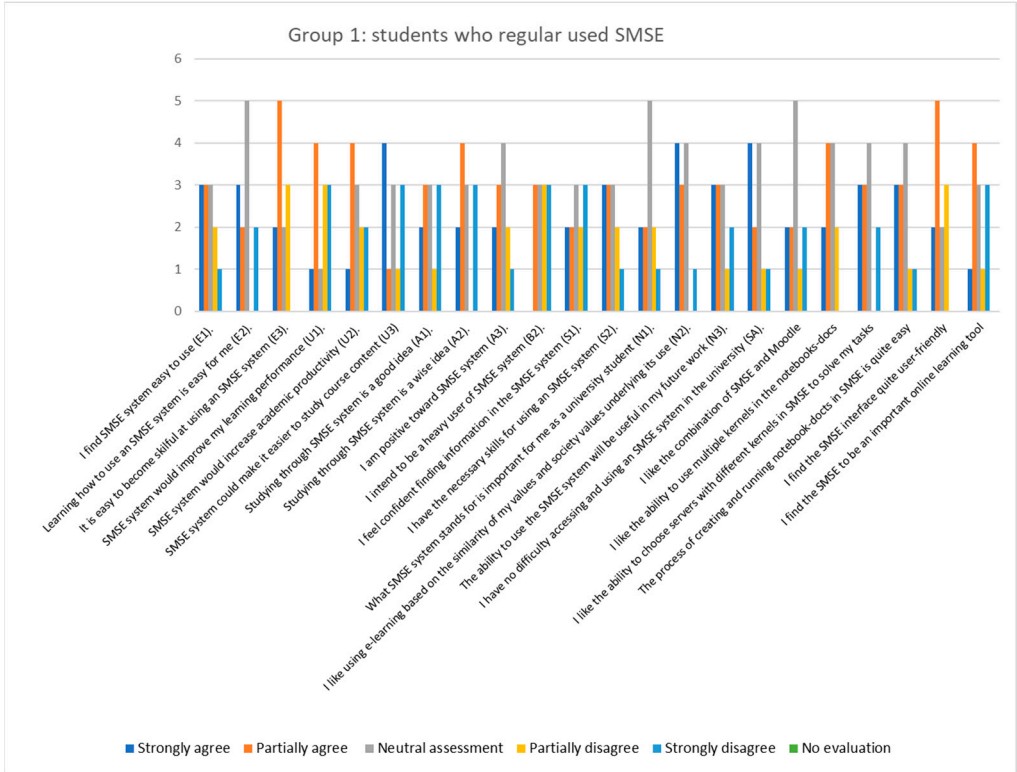

**Figure 13.** The answers of Group 1 students who regularly used SMSE—histogram of cumulative results.

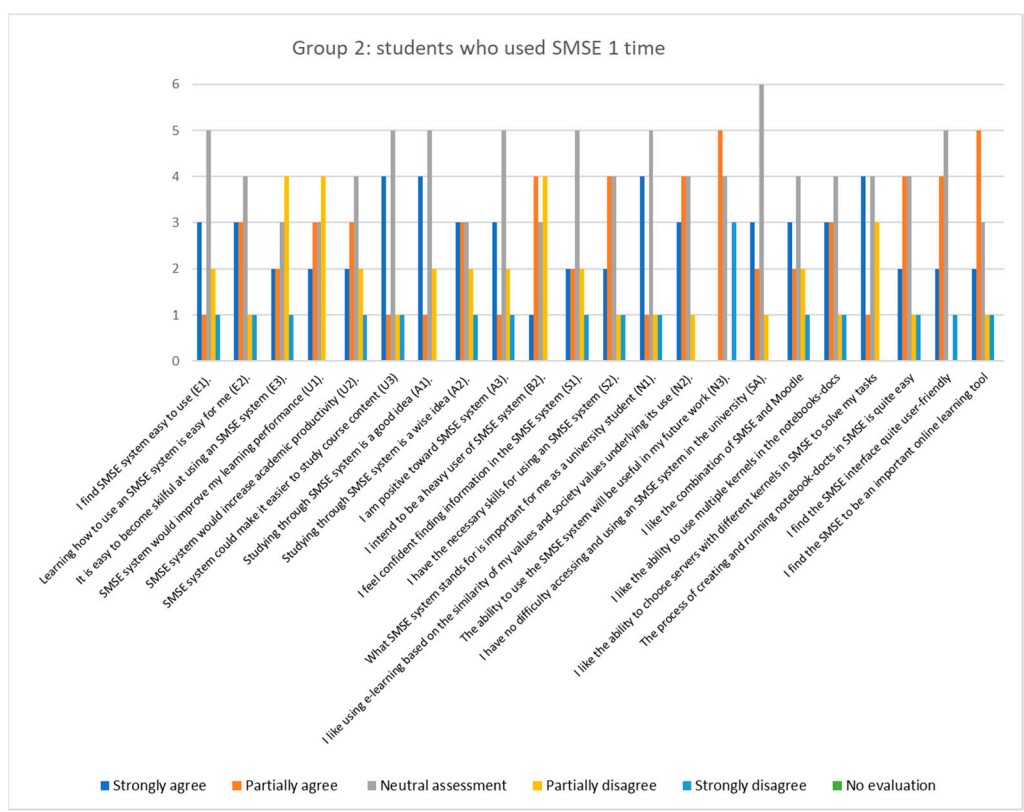

**Figure 14.** The answers of Group 2 students who recently started to use SMSE—histogram of cumulative results.

In addition, we received several comments and suggestions concerning the use of the SMSE. For example, "When the presentation of this system was made, I did not really understand the idea of combining the report and the code, that is, compiling the code in the report, in my opinion, finding errors, if the code is scattered throughout the file, will be quite difficult and take more attention. In this case, it does not increase productivity, but the very idea of a personal laboratory, in general, if there is more information about the use of this system and its interface, it can facilitate the use of this system and learning".

To understand how much the perception of students in both groups differs and what the trend is in changing their minds, we analyzed the answers of both groups in five categories: "Strongly agree", "Partially agree", "Neutral assessment", "Partially disagree" and "Strongly disagree". The summary of answers is depicted in Figure 15, "Cumulative results of a survey of both students' groups", which shows the number of answers in each of these five categories.

When we consider two groups of students in light of the transformation of their perceptions, we see that the number of "Neutral assessment" answers decreased dramatically after two semesters when the SMSE was used in the learning process. This demonstrates that the students had a more pronounced perception of the platform. Simultaneously, the number of negative marks increased almost twice (answers "Strongly disagree"), which indicates that a particular number of students met problems in the acquisition of the new tool. In addition, a considerable growth in "Partially agree" shows considerable progress in the acquisition of the SMSE platform.

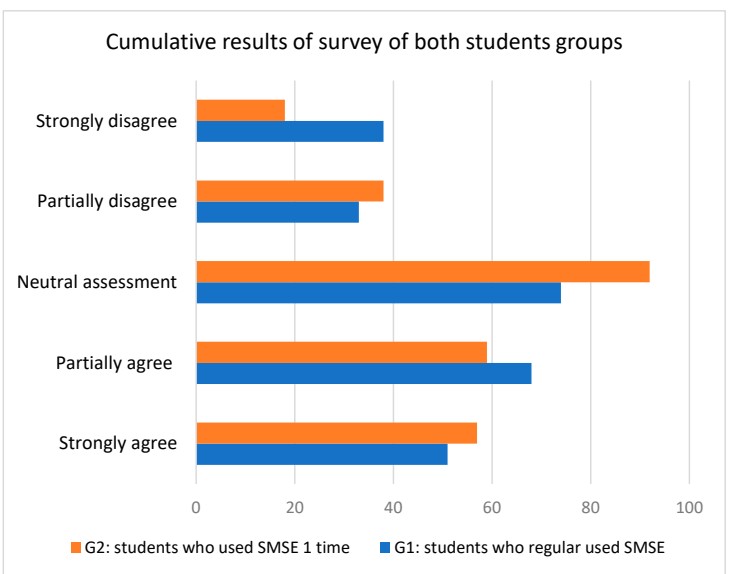

**Figure 15.** Cumulative results of a survey of both student groups.

*5.2. Discussions*

Table 1 provides a consolidated view of the answers of the students from Group 1 and Group 2. It is clear that the means of the evaluation marks of both groups are rather similar: 2.7689 for Group 1 and 2.6250 for Group 2. However, the average perception of students in Group 1 is more positive than the average perception of students in Group 2. The standard deviations in the values of the responses in both groups are rather similar (standard deviation $SD_1$ is equal to 1.3079 in Group 1 against $SD_2$ that is equal to 1.1979 in Group 2, indicating that the larger personal experiences of the students lead to more divergent evaluations).

We have ranked the answers in both groups, aiming to understand which benefits of the platform might be more important for students in both groups. The willingness to be "a heavy user of the SMSE system" (topic 10) is recognized in both groups: first order in Group 1 and second order in Group 2. Topic 11: "I feel confident finding information in the SMSE system" was ranked in third place in Group 2 and also in third place in Group 1. It is worth noting that the majority of evaluation topics have been evaluated differently, which reveals a transformation in students' perceptions during the practical learning process when the SMSE platform is applied.

To evaluate the transformation of students' perceptions, we calculated the difference in evaluation marks given by the students in Group 1 and Group 2. More precisely, we calculated the difference between the means for each of the 22 topics in percentage. We noted the growth of positive perceptions, which was derived from several months of experience using the platform (answers to questions 4, 7, 10 and 22). Thus, the students revealed that the SMSE system would improve their learning performance; they agreed that studying through the SMSE system is a good idea; they intended to be heavy users of the platform; and they find the platform as an important online learning tool.

The growth of negative perceptions concerning the SMSE platform as a learning tool appeared due to the answers to questions 1, 3, 15, 20. This indicates the difficulties students face in acquiring the new tool. In addition, the answers point out the students' doubts about the application of their skills in SMSE in their future work (topic 15). Even if this is not directly related to the current educational process, it might be considered as a demotivating factor, considering the uselessness of such knowledge for a student in the future. Therefore, it would be worth, in the student's training, to evaluate the life cycle of the platform beyond the students' course and to suggest examples from real life of useful applications of the platform. This approach might increase the motivation of the students.

**Table 1.** Consolidated answers of the students from Group 1 and Group 2.

| No | The Topic | Group 1: Students Who Regularly Used SMSE | | | | Group 2: Students Who Used SMSE 1 Time | | | | Diff. % | Order |
|---|---|---|---|---|---|---|---|---|---|---|---|
| | | N | Mean | SD | Order | N | Mean | SD | Order | | |
| *1* | *2* | *3* | *4* | *5* | *6* | *7* | *8* | *9* | *10* | *11* | *12* |
| 1 | I find SMSE system easy to use (PE1). | 12 | 2.5833 | 1.3114 | 14 | 12 | 2.7500 | 1.2881 | 5 | −6.45 | 20 |
| 2 | Learning how to use an SMSE system is easy for me (PE2). | 12 | 2.6667 | 1.3707 | 12 | 12 | 2.5000 | 1.2432 | 13 | 6.25 | 11 |
| 3 | It is easy to become skillful at using an SMSE system (PE3). | 12 | 2.5000 | 1.0871 | 17 | 12 | 3.0000 | 1.2792 | 2 | −20.0 | 22 |
| 4 | SMSE system would improve my learning performance (PU1). | 12 | 3.2500 | 1.4222 | 2 | 12 | 2.7500 | 1.1382 | 5 | 15.38 | 4 |
| 5 | SMSE system would increase academic productivity (PU2). | 12 | 3.0000 | 1.2792 | 5 | 12 | 2.7500 | 1.2154 | 5 | 8.33 | 10 |
| 6 | SMSE system could make it easier to study course content (PU3) | 12 | 2.8333 | 1.6422 | 8 | 12 | 2.5000 | 1.3143 | 13 | 11.76 | 5 |
| 7 | Studying through SMSE system is a good idea (AT1). | 12 | 3.0000 | 1.4771 | 5 | 12 | 2.4167 | 1.1645 | 20 | 19.44 | 1 |
| 8 | Studying through SMSE system is a wise idea (A2). | 12 | 2.8333 | 1.4668 | 8 | 12 | 2.5833 | 1.3114 | 10 | 8.82 | 8 |
| 9 | I am positive toward SMSE system (AT3). | 12 | 2.7500 | 1.2154 | 11 | 12 | 2.7500 | 1.2881 | 5 | 0.00 | 13 |
| 10 | I intend to be a heavy user of SMSE system (BI2). | 12 | 3.5000 | 1.1677 | 1 | 12 | 2.8333 | 1.0299 | 3 | 19.05 | 2 |
| 11 | I feel confident finding information in the SMSE system (SE1). | 12 | 3.1667 | 1.4668 | 3 | 12 | 2.8333 | 1.1934 | 3 | 10.53 | 7 |
| 12 | I have the necessary skills for using an SMSE system (SE2). | 12 | 2.5833 | 1.3114 | 14 | 12 | 2.5833 | 1.1645 | 10 | 0.00 | 13 |
| 13 | What SMSE system stands for is important for me as a university student (SN1). | 12 | 2.8333 | 1.1934 | 8 | 12 | 2.5000 | 1.3143 | 13 | 11.76 | 5 |
| 14 | I like using e-learning based on the similarity of my values and society values underlying its use (SN2). | 12 | 2.2500 | 1.2154 | 22 | 12 | 2.2500 | 0.9653 | 22 | 0.00 | 13 |
| 15 | The ability to use the SMSE system will be useful in my future work (SN3). | 12 | 2.6667 | 1.4355 | 12 | 12 | 3.0833 | 1.2401 | 1 | −15.6 | 21 |
| 16 | I have no difficulty accessing and using an SMSE system in the university (SA). | 12 | 2.4167 | 1.3114 | 21 | 12 | 2.4167 | 0.9962 | 20 | 0.00 | 13 |
| 17 | I like the combination of SMSE and Moodle | 12 | 2.9167 | 1.3114 | 7 | 12 | 2.6667 | 1.3027 | 9 | 8.57 | 9 |
| 18 | I like the ability to use multiple kernels in the notebooks-docs | 12 | 2.5000 | 1.0000 | 17 | 12 | 2.5000 | 1.2432 | 13 | 0.00 | 13 |
| 19 | I like the ability to choose servers with different kernels in SMSE to solve my tasks | 12 | 2.5833 | 1.3790 | 14 | 12 | 2.5000 | 1.2432 | 13 | 3.23 | 12 |
| 20 | The process of creating and running notebook-docs in SMSE is quite easy | 12 | 2.5000 | 1.2432 | 17 | 12 | 2.5833 | 1.1645 | 10 | −3.33 | 19 |
| 21 | I find the SMSE interface quite user-friendly | 12 | 2.5000 | 1.0871 | 17 | 12 | 2.5000 | 1.0871 | 13 | 0.00 | 13 |
| 22 | I find the SMSE to be an important online learning tool | 12 | 3.0833 | 1.3790 | 4 | 12 | 2.5000 | 1.1677 | 13 | 18.92 | 3 |
| | Average: | | 2.7689 | 1.3079 | | | 2.6250 | 1.1979 | | | |

In our research, we evaluated the correlation between particular topics of the Technology Acceptance Model and the overall evaluation of the SMSE platform. In particular, we evaluated how a topic like "Studying through the SMSE system is a good idea" (topic 7) as one of the students' attitude indicators might correlate with the overall evaluation topic. A

rather strong correlation coefficient of 0.9371 for Group 1 and 0.9312 for Group 2 confirmed this strong correlation between this topic and the overall evaluation of the SMSE platform.

On the other hand, students in Group 1 and Group 2 demonstrated a rather different perception that was implied by divergent correlation coefficients. For example, the correlation between topic 15 "The ability to use the SMSE system will be useful in my future work" and the overall evaluation topic for Group 1 was 0.3674. However, the correlation for the same topic in Group 2 (beginners in the usage of the platform) was much higher: 0.7044. In addition, the correlation between topic 12 ("I have the necessary skills for using an SMSE system") and the overall evaluation topic for Group 1 was 0.3715. However, the correlation for Group 2 (beginners using the platform) was much higher: 0.8235.

For both topics, the correlation coefficient was larger in Group 2 ("beginners"). We could interpret these results in such a way that more experienced students, to a lesser extent, evaluate the new learning platform with the opportunity to use it beyond the school courses. It also looks like even experienced students evaluate their skills rather equally to less experienced students; however, their perceptions of their skills show less correlation with the overall evaluation of the new learning platform.

## 6. Conclusions

The Shared Modeling and Simulation Environment project created a web-based, multi-functional e-learning environment that combines a variety of tools and technologies aiming to give students the possibility to be involved in all sorts of teaching activities, such as lectures, exercises, simulations, etc., through the use of one single online environment. This platform provides a flexible and highly customizable solution to merge the online learning platform Moodle and the on-demand preconfigured working environments of Jupyter Notebooks in an efficient way. This approach provides an enhanced and highly interactive learning process of STEM subjects for both in-class and remote learning programs as well as courses. The SMSE platform may come in handy for sustainable teaching and learning in a period of instability associated with pandemics such as COVID-19, and Russia's military aggression, providing an opportunity for students and teachers who were forced to become displaced persons to continue their studies and jobs.

We are aware of the restrictions of the research since the survey of only two quite small groups of students cannot guarantee an exhaustive perception of the newly developed courses and teaching/learning methods. Indeed, we were only able to test a rather small number of students.

For future research, improvements should be made to the questionnaire form for the assessment of students' feedback. Therefore, it looks reasonable to supplement the questionnaire with additional open-ended questions, which should encourage students to make more comments concerning possible improvements in the content of the courses and the teaching methods. We are also interested in evaluating to what extent the involvement of industry representatives in the teachers' team would increase the motivation of the students.

We plan to continue the development of the SMSE platform in the frame of a new ERASMUS+ project, DIGITRANS, which was started in December 2023. Our future strategy on expanding the usage of SMSE will be based on the creation of a set of virtual servers for the acquisition of new courses with practical training in combination with the programming of remote hardware devices.

**Author Contributions:** Conceptualization, A.Z. and V.K.; methodology, A.Z. and V.K.; software, O.D.; validation, S.V. and L.E.; formal analysis, V.K.; investigation, O.D. and A.Z.; resources, V.K.; data curation, O.D.; writing—original draft preparation, V.K.; writing—review and editing, R.B.; visualization, V.K.; supervision, A.Z. All authors have read and agreed to the published version of the manuscript.

**Funding:** This research received no external funding.

**Institutional Review Board Statement:** Not applicable.

**Informed Consent Statement:** Not applicable.

**Data Availability Statement:** Data are contained within the article.

**Acknowledgments:** The authors of this research are grateful to the students and academic staff of the Chernihiv Polytechnic National University, who supported this research work by participating in the survey and by giving useful advises.

**Conflicts of Interest:** The authors declare no conflicts of interest.

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
