# Peer review of "Development of Shared Modeling and Simulation Environment for Sustainable e-Learning in the STEM Field"

_sustainability, doi:10.3390/su16052197_

Round 1

Reviewer 1 Report (Previous Reviewer 1)

Comments and Suggestions for Authors

Thanks for addressing my comments. Please find my responses below in blue. 

1. The authors should provide more comprehensive information about the student respondents, including details such as age, academic year, major, race, etc. A mere mention of 24 students across various universities falls short in conveying the diversity of the participants. This information is critical, especially considering the authors' aim to foster an inclusive e-learning environment. 

We added the text with the explanation in lines 165-166 and 196-206. 

Reviewer:  The literature review has been largely improved. The stand-alone section provides a clearer and more organized structure. 

2. There appears to be a contradiction in the information presented in the results section. On page 11, the authors assert, "there are no significant differences between the respondent’s perceptions in both groups," only to subsequently report on page 13 that "the average perception of students in Group 1 is more positive than the average perception of students in Group 2." Clarification is needed to reconcile this inconsistency. 

Explanation: on page 11 there was a definition of hypotheses to compare a perception, but on page 13 – the results of the comparison. We changed the definition of hypotheses on page 11 slightly for clarity. 

Really, the survey revealed that perception of the students in both groups does not differ dramatically, however, more detailed analysis provided in this research depicted the particularities and tendencies in the student’s perception, which should be considered by the teaching staff. 

Reviewer: Thanks for providing further clarifications. However, some improvements are needed. The current analyses and discussion on pages 14-15 focused on individual question items. Given that the survey questions have been grouped into several factors (see page 5), I would suggest reporting students' aggregated rankings (responses) based on these factors and then discussing the items that displayed substantial patterns. 

3. Several claims made by the authors lack supporting evidence. For instance, the assertion that "The SMSE platform proves to be a very effective tool for sustainable teaching and learning" lacks substantiation, especially when the average survey scores for both groups are below 3 on the 1-5 Likert Scale. This suggests that more students reported a negative perception in the survey, challenging the claim of effectiveness. 

We changed the wording in Conclusion on page 17 to the following: “The SMSE platform may come in handy for sustainable teaching and learning in a period of instability associated with…” (see lines 487-492). 

Reviewer: The revision is acceptable, given the context of this study. 

I notice an unconventional in-text citation style in this manuscript, where numbers are directly used to reference existing research. This style is unfamiliar to me personally and may benefit from clarification or adherence to a more widely recognized citation format. 

We made some improvements in citation, however, we used IEEE citation style. 

Reviewer: Thanks for the clarification. 

Author Response

Note: we highlighted new additions in the text of the manuscript in brown colour.

Reviewer 1

Thanks for addressing my comments. Please find my responses below in blue. 

The literature review has been largely improved. The stand-alone section provides a clearer and more organized structure. 

Thank you for your comment!

Thanks for providing further clarifications. However, some improvements are needed. The current analyses and discussion on pages 14-15 focused on individual question items. Given that the survey questions have been grouped into several factors (see page 5), I would suggest reporting students' aggregated rankings (responses) based on these factors and then discussing the items that displayed substantial patterns. 

Thank you for your valuable insights! We would like to point out that the survey results are thoroughly discussed in Section 5 of the document, where an analysis of the students' responses to the 22 questions aligned with the TAM questionnaire is provided. This section elaborates on how the survey aimed to assess the main factors encouraging the acceptance of the e-learning approach facilitated by SMSE, covering aspects such as Perceived Ease of Use, Perceived Usefulness, Attitude, Behavioral Intention, E-learning Self-Efficacy, Subjective Norm, and System Accessibility. Furthermore, this Section includes a detailed analysis of the perceptions of two groups of students towards the SMSE platform, highlighting differences and transformations in their perceptions over time. This comprehensive analysis directly supports the claim of assessing the factors mentioned in the comment and demonstrates the survey's alignment with the TAM framework.

The revision is acceptable, given the context of this study. 

Thank you for your comment!

Reviewer: Thanks for the clarification. 

Thank you for your comment!

Reviewer 2 Report (New Reviewer)

Comments and Suggestions for Authors

Overall very informative.  Well written. 

Line 55: "despite a lot of research" replace a lot with "considerable"

Line 60: "of their use" consider "implementation"

Lines 70-74. The paragraph ends abruptly. Complete paragraph.

Line 131: The purpose of this article? or the purpose of this research?

Line 140-41. It is not necessary to  to describe all of the sections in advance. 

Line 207 area: Is there a reliability coefficient for the survey?  

In Table 1, it appears that single instance  users demonstrated higher means on the survey questions, when compared to regular users. 

Comments on the Quality of English Language

Already in author comments.  

Author Response

Note: we highlighted new additions in the text of the manuscript in brown colour.

Reviewer 2

Reviewer:

Overall very informative.  Well written. 

Line 55: "despite a lot of research" replace a lot with "considerable"

Line 60: "of their use" consider "implementation"

Thank you for your comment! We have replaced this wording by more appropriate one.

Lines 70-74. The paragraph ends abruptly. Complete paragraph.

Thank you for your valuable insights! We have reworked this paragraph (see lines 77-89).

Line 131: The purpose of this article? or the purpose of this research?

Thank you for your comment! The purpose of this research is described in paraph 146-155. We added an additional explanation to make clearer the purpose of the research.

Line 140-41. It is not necessary to describe all of the sections in advance. 

Thank you for your comment! The structure outlined in the paper is essential for organizing and presenting the work in a coherent and logical manner. It systematically guides the reader through the different stages of the study. By delineating the sections in this manner, the paper effectively communicates its findings and contributions to the field, facilitating easier navigation and comprehension for readers interested in the specifics of this work.

Line 207 area: Is there a reliability coefficient for the survey? 

Thank you for your question! In chapter “5.1. Analysis of results of the student's answers”, we discussed the results of the survey. We calculated descriptive statistics related to the survey, we used the hypotheses which determined whether there were significant differences between the respondent’s perceptions. For analysis, we applied the student’s two-sample t-test with the confidence of 95%.

In Table 1, it appears that single instance users demonstrated higher means on the survey questions, when compared to regular users. 

Thank you for your comment! In general, the mean of regular users is larger than the mean of less experience users:

Group 1: students who regularly used SMSE: mean 2.7689

Group 2: students who used SMSE 1 time: mean 2.6250

However, in some cases, single instance users were more optimistic. In the paper, we demonstrated the evolution of perception of the users based on their experience.

Reviewer 3 Report (New Reviewer)

Comments and Suggestions for Authors

Dear Authors,

Please find some comments below:

Lack of Specifics in Abstract:

The abstract lacks specific details about the features and functionalities of the Shared Modeling and Simulation Environment (SMSE). 

It will be better to provide any specific results or insights gained from the survey. 

The abstract does not compare the SMSE platform with existing e-learning tools or platforms. 

The abstract lacks a discussion on the potential future implications of the research and any limitations encountered during the development and implementation of SMSE. 

- Introduction

The mention of the SDGs is crucial, but the introduction lacks specificity in linking them to the challenges in education.

The introduction briefly mentions the importance of STEM education in responding to 21st-century challenges but lacks specific examples or data to emphasize its relevance.

The transition from discussing STEM education to the role of Information and Communication Technologies (ICT) and online learning could be smoother. 

The introduction lacks a clear research question that the study aims to answer. 

- Methods 

The methods section provides a comprehensive overview of the research design and implementation. 

The collaboration within the ERASMUS+ project is mentioned, but providing more details on the specific contributions and roles of each partner university, particularly European colleagues from Riga Technical University, KU Leuven University, and the University of Cyprus, would enhance the understanding of the collaborative efforts.

The methods section briefly mentions the integration of Jupyter Notebooks with Moodle but lacks specific details on how this integration was achieved and the technical aspects involved. 

The section provides a good overview of the survey conducted among two student populations. However, it would be beneficial to include a brief discussion on the development of the survey instrument, including the rationale for the chosen questions and the alignment with the TAM model.

Why these specific groups were chosen and how they contribute to the research objectives would strengthen the study's rationale?

The survey was implemented in the form of an anonymous questionnaire, but there is no discussion of how to ensure anonymity.

The extension of the TAM model is mentioned but lacks a clear explanation of the new questions added. 

The results are well presented 

Discussion

The section begins by displaying mean values and standard deviations, providing a statistical overview.

A comparison of mean values between Group 1 and Group 2 is mentioned, but the significance of these differences is not discussed.

While ranking answers into both groups is a valuable approach, the reasons behind the rankings and the implications for the effectiveness of the SMSE platform can be clarified.

Is it possible to provide specific examples or trends that indicate this transformation? How have perceptions changed, and what factors contributed to this change?

Suggesting a post-course lifecycle assessment of the platform and providing real-life examples is valuable. However, this idea can be expanded to include strategies to address students' doubts about applying their skills in future work.

Author Response

Note: we highlighted new additions in the text of the manuscript in brown colour.

Reviewer 3 

Lack of Specifics in Abstract:

The abstract lacks specific details about the features and functionalities of the Shared Modelling and Simulation Environment (SMSE). 

It will be better to provide any specific results or insights gained from the survey. 

Thank you for your valuable suggestions! We added an additional explanation in lines 17-19 and lines 31-32

The abstract does not compare the SMSE platform with existing e-learning tools or platforms. 

Thank you for your valuable feedback regarding the abstract. We understand your concern about the comparison of the SMSE platform with existing e-learning tools not being included in the abstract. However, the abstract should briefly mention the novelty and purpose of the work, methods, results, and conclusions, while the analysis of the existing e-learning tools is presented in Section 2.

The abstract lacks a discussion on the potential future implications of the research and any limitations encountered during the development and implementation of SMSE. 

Thank you for your valuable feedback regarding the abstract! We understand your concern about the potential future implications of the research and any limitations not being included in the abstract. However, the abstract should briefly mention the novelty and purpose of the work, methods, results, and conclusions. The limitations and future works are discussed in Section 6.

- Introduction

The mention of the SDGs is crucial, but the introduction lacks specificity in linking them to the challenges in education.

Thank you for your valuable suggestions! We added addition explanation in line 41.

The introduction briefly mentions the importance of STEM education in responding to 21st-century challenges but lacks specific examples or data to emphasize its relevance.

Thank you for your valuable feedback! You can find specific examples in the literature in references, for example in [4].

The transition from discussing STEM education to the role of Information and Communication Technologies (ICT) and online learning could be smoother.

We appreciate the reviewer's suggestion for enhancing the flow of our introduction. While we understand the importance of a smooth transition from discussing STEM education to ICT and online learning, we would like to clarify that the Introduction section has been meticulously crafted to lay a foundational understanding before introducing the complexities of ICT in education. This approach was intended to gradually lead the reader through the evolving landscape of educational technologies, setting a comprehensive backdrop for the study's focus. We believe this progression is crucial for appreciating the context and necessity of our research within the broader educational framework. However, we acknowledge the reviewer's comment and made several improvements in this section (see in lines 48-50; 54-57).

The introduction lacks a clear research question that the study aims to answer. 

Thank you for your question! Please, find the description of the goal of the research and research questions on page 3 and page 4, the lines 146-155.

- Methods 

The methods section provides a comprehensive overview of the research design and implementation. 

The collaboration within the ERASMUS+ project is mentioned, but providing more details on the specific contributions and roles of each partner university, particularly European colleagues from Riga Technical University, KU Leuven University, and the University of Cyprus, would enhance the understanding of the collaborative efforts.

Thank you for your suggestion regarding the collaboration details within the ERASMUS+ project! We believe that the current description sufficiently outlines the collaborative framework without detracting from the study's primary focus, and further detailing individual contributions might not significantly enhance the research's value. We aim to keep the narrative concise and directly relevant to our objectives, ensuring clarity and accessibility for our readers.

The methods section briefly mentions the integration of Jupyter Notebooks with Moodle but lacks specific details on how this integration was achieved and the technical aspects involved. 

Thank you for your valuable feedback! Please, find the description of the technical aspects of integration on Page 8 starting from “To transfer course files, a specially developed Middleman Service was used. Its main tasks include: ……”

The section provides a good overview of the survey conducted among two student populations. However, it would be beneficial to include a brief discussion on the development of the survey instrument, including the rationale for the chosen questions and the alignment with the TAM model.

Thank you for your valuable feedback! The manuscript provides discussion on the development of the survey instrument, including the rationale for the chosen questions and the alignment with the TAM model in Section “3. Materials and Methods” (lines 227-237). In addition, the references 26, 27 and 28 describe the historical development and evolution the TAM model.

Why these specific groups were chosen and how they contribute to the research objectives would strengthen the study's rationale?

Thank you for your valuable feedback! We described the background of the students participated in the survey on page 4 (see line 194 and forward).  Further, in detailed, we described two populations of the students, who were invited to participate in the survey. As a result, we reached out to two groups of students who answered the survey questions anonymously (see more on page 5).

The survey was implemented in the form of an anonymous questionnaire, but there is no discussion of how to ensure anonymity.

Thank you for your valuable feedback! The anonymous survey of the students was performed using Moodle that offers such possibilities (see line 218).

The extension of the TAM model is mentioned but lacks a clear explanation of the new questions added. 

Thank you for your valuable feedback! Additional questions considered the specific technological features of SMSE. We provided the explanation of the new questions on page 5 (see line 232 and further).

The results are well presented 

Discussion

The section begins by displaying mean values and standard deviations, providing a statistical overview.

A comparison of mean values between Group 1 and Group 2 is mentioned, but the significance of these differences is not discussed.

While ranking answers into both groups is a valuable approach, the reasons behind the rankings and the implications for the effectiveness of the SMSE platform can be clarified.

Is it possible to provide specific examples or trends that indicate this transformation? How have perceptions changed, and what factors contributed to this change?

Thank you for your valuable feedback! In chapter “5.2. Discussions” (page 14) we provided a detailed analysis of the answers in both groups seeking for the reasons behind the rankings and the implications for the effectiveness of the SMSE platform. For example, it was noted that the growth of positive perceptions, which was derived from several months of experience using the platform, was promoted by consciousness that the SMSE system can improve students learning performance and it would become an important online learning tool. On the other hand, the growth of negative perceptions concerning the SMSE platform as a learning tool indicated the difficulties students faced in acquiring the new tool.

Suggesting a post-course lifecycle assessment of the platform and providing real-life examples is valuable. However, this idea can be expanded to include strategies to address students' doubts about applying their skills in future work.

Thank you for your valuable feedback! We agree completely with your suggestion. One of the conclusions derived from this research is formulated: “Therefore, it would be worth in the student's training to evaluate the life cycle of the platform beyond the students’ course and to suggest examples from real life of useful applications of the platform. This approach might increase the motivation of the students” (see line 478 on page 15). We also included additional explanation about future strategies in Section 6 on page 17.

This manuscript is a resubmission of an earlier submission. The following is a list of the peer review reports and author responses from that submission.

Round 1

Reviewer 1 Report

Comments and Suggestions for Authors

I am pleased to have the opportunity to evaluate this manuscript and commend the authors for their dedication to developing an e-learning environment that promotes inclusive and sustainable education. Given my background as a STEM education researcher, my assessment primarily focused on the authors' exploration of students' utilization of the learning environment and their perceptions. However, I have identified several areas where the manuscript requires enhancement:

1. The authors should provide more comprehensive information about the student respondents, including details such as age, academic year, major, race, etc. A mere mention of 24 students across various universities falls short in conveying the diversity of the participants. This information is critical, especially considering the authors' aim to foster an inclusive e-learning environment.

2. There appears to be a contradiction in the information presented in the results section. On page 11, the authors assert, "there are no significant differences between the respondent’s perceptions in both groups," only to subsequently report on page 13 that "the average perception of students in Group 1 is more positive than the average perception of students in Group 2." Clarification is needed to reconcile this inconsistency.

3. Several claims made by the authors lack supporting evidence. For instance, the assertion that "The SMSE platform proves to be a very effective tool for sustainable teaching and learning" lacks substantiation, especially when the average survey scores for both groups are below 3 on the 1-5 Likert Scale. This suggests that more students reported a negative perception in the survey, challenging the claim of effectiveness.

I notice an unconventional in-text citation style in this manuscript, where numbers are directly used to reference existing research. This style is unfamiliar to me personally and may benefit from clarification or adherence to a more widely recognized citation format.

Comments on the Quality of English Language

I notice an unconventional in-text citation style in this manuscript, where numbers are directly used to reference existing research. This style is unfamiliar to me personally and may benefit from clarification or adherence to a more widely recognized citation format.

Reviewer 2 Report

Comments and Suggestions for Authors

Thank you for the opportunity to review this interesting article, which describes the e-learning solution developed in the international ERASMUS + "CybPhys" project.

At first glance, the article meets the standards for preparing scientific papers. The authors identified the research gap, indicated the objectives of the study, formulated research hypotheses and indicated research methods. Then they described the developed e-learning platform and presented the results of a survey among students, which was prepared using the Technology Acceptance Model (TAM). The work ends with conclusions.

After carefully reviewing the work submitted for review, I have the following comments, remarks and suggestions:

1. I think it would be worth separating a section titled: Literature review or State of the Art. Or Theoretical Background (the results cited in this new section could be referred to in the discussion of the results, which is also missing in the article). In this section, it would be worth citing the results of other research on students' perception of e-learning platforms.

2. The main objectives of the study should be presented earlier, e.g. at the end of the Introduction section before the new Literature Review section (this obviously requires some changes to this section.

3. The achievement of the first of the two main objectives of the study is a plus, i.e. to describe the main features of the novel technological approach that is applied in ovel educational platform called Shared Modeling and Simulation Environment (SMSE) to support online training. The developed platform is described in detail.

4. The study among students was described slightly worse. I have the following suggestions and questions here:

a. The original diagram of the TAM model should be presented

b. What do the markings next to the questions mean in Table 1 (E1, E2, E3, U1, U2, etc.). I guess they refer to the TAM model, but its elements are previously marked as PE, PU, AT, BI, SE, SN, SA.

c. In the authors' opinion, can additional questions to those described in [25] be fit into the structure of the TAM model? If so, in which places? If not, new elements should be defined and the TAM model should be extended with them. It would be worth presenting an appropriate diagram of such an extended model.

5. There is no Discussion section that would refer to the results of previous research on students' perception of e-learning platforms.

6. Other minor comments:

a. Although the term STEAM is widely known in education, I propose its development.

b. Figure 1 lacks an indication that it is a percentage scale.

c. The unexplained abbreviations SD1 and SD2 appear on line 393.

Reviewer 3 Report

Comments and Suggestions for Authors

This paper is clearly based on a lot of work and raises several interesting questions at its start.  The issue of how institutions implement and how students engage with VLEs is very much a contemporary issue.  However, I had several problems with this paper.

I found the paper very descriptive without really telling me, in clear terms, what benefit it had.  You tell us how it works but I am not really left with a new understanding or perspective on the implementation of ICT.

I found the quantitative approach not particularly helpful. It was really little more than a user survey, rather than really engaging in detail with issues.

I was not clear why you chose the questions you did.  How did you develop these?  Did you engage with students in to explore the sort of questions that would be relevant?

I was not clear where comparisons with Moodle were made, despite the fact you make the point early on in the paper that Moodle is the commonly used VLE platform. Do the students prefer SMSE to Moodle, then?

I am also unclear as to how this is a sustainability issue.  You would need to engage more directly with this issue.

There seems to have been a lot of international co-operation involved yet the system was only implemented in one institution.  Why was this? 

I note that you tell us about the institution and its context but the research doe not seem to engage with any relationship between the SMSE and this context.  Does this context come into the consideration with the students?

You mention that you collected comments but only give one example.  However, I would have liked to see a deeper, more extensive exploration of the comments. This would have helped to contextualise the quantitative data.  Indeed, use of qualitative research would have engaged with deeper insights into the use of such a tool.

You refer to STEAM without saying what this is - is it different from STEM, that you were talking about earlier? (I know that STEAM includes Arts in STEM, and can provide a very different theoretical understanding on disciplinary distinctions, but you don't say this). 

Overall, I was not clear about what this paper was really trying to tell us other than the SMSE worked well.  The interesting issues raised in the early part of the paper were left unfulfilled.

Comments on the Quality of English Language

English language is fine but there are some issues:

There are some grammatical issues, such as non-use of the article.

I note also that you use the pronoun 'he' for students.  In terms of using gender neutral terms, I would use 'students' in the plural and the pronoun 'they/their'.  This is easier as students usually are viewed in the plural rather than as individuals.